# Genomic Landscape Highlights Molecular Mechanisms Involved in Silicate Solubilization, Stress Tolerance, and Potential Growth-Promoting Activity of Bacterium *Enterobacter* sp. LR6

**DOI:** 10.3390/cells11223622

**Published:** 2022-11-15

**Authors:** Gaurav Raturi, Yogesh Sharma, Rushil Mandlik, Surbhi Kumawat, Nitika Rana, Hena Dhar, Durgesh Kumar Tripathi, Humira Sonah, Tilak Raj Sharma, Rupesh Deshmukh

**Affiliations:** 1Department of Agriculture Biotechnology, National Agri-Food Biotechnology Institute (NABI), Mohali 140306, India; 2Department of Biotechnology, Panjab University, Chandigarh 160014, India; 3Crop Nano Biology and Molecular Stress Physiology Lab, Amity Institute of Organic Agriculture, Amity University Uttar Pradesh, Noida 201313, India; 4Division of Crop Science, Indian Council of Agricultural Research, New Delhi 110001, India; 5Center for Digital Agriculture, Plaksha University, Mohali 140306, India

**Keywords:** *Enterobacter*, genome sequencing, plant-growth-promoting bacteria, GlpF, metalloid transport

## Abstract

Silicon (Si) is gaining widespread attention due to its prophylactic activity to protect plants under stress conditions. Despite Si’s abundance in the earth’s crust, most soils do not have enough soluble Si for plants to absorb. In the present study, a silicate-solubilizing bacterium, *Enterobacter* sp. LR6, was isolated from the rhizospheric soil of rice and subsequently characterized through whole-genome sequencing. The size of the LR6 genome is 5.2 Mb with a GC content of 54.9% and 5182 protein-coding genes. In taxogenomic terms, it is similar to *E. hormaechei* subsp. xiangfangensis based on average nucleotide identity (ANI) and digital DNA–DNA hybridization (dDDH). LR6 genomic data provided insight into potential genes involved in stress response, secondary metabolite production, and growth promotion. The LR6 genome contains two aquaporins, of which the aquaglyceroporin (GlpF) is responsible for the uptake of metalloids including arsenic (As) and antimony (Sb). The yeast survivability assay confirmed the metalloid transport activity of GlpF. As a biofertilizer, LR6 isolate has a great deal of tolerance to high temperatures (45 °C), salinity (7%), and acidic environments (pH 9). Most importantly, the present study provides an understanding of plant-growth-promoting activity of the silicate-solubilizing bacterium, its adaptation to various stresses, and its uptake of different metalloids including As, Ge, and Si.

## 1. Introduction

Silicon (Si), one of the most abundant elements on earth’s crust, is gaining wide attention due to an increasing number of reports showing its benefits for plant growth. Although Si is not an essential element, it provides numerous benefits to plants, particularly under stress conditions, and is thus considered a quasi-essential element. Si amendment can increase the carbon sequestration to the rice panicle and increase the photo-assimilation of carbon in plants [1]. Although Si is abundant in the soil, it is mostly in the insoluble silicate form, which cannot be absorbed by the plant roots [2,3]. Insoluble silicates present in soil can be converted into soluble forms by the various chemical, physical, and biological activities of plants and microorganisms. Bacterial interventions have been reported to accelerate solubilization [4,5]. The silicate-solubilizing bacteria (SSB) can convert the insoluble silicate into orthosilicic acid (H_4_SiO_4_), a soluble form of Si that can be easily uptaken by the plants along with the water. Once absorbed, the Si is transported to active growing parts of the plant, makes complexes with the carbohydrates, sugar derivatives, and hydroxy groups of proteins involved in signal transduction. Therefore, Si can interact with a number of key components involved in the signaling system for plant stress, provide resistance to abiotic stress, strengthen the cell wall, inhibit fungal infection due to rapid production of defense compounds, and improve the overall growth and development of the plants [6,7]. Uptake of Si from the soil solution to the plant root is facilitated with the help of aquaporins (AQPs). The AQPs, a class of channel-forming proteins belonging to the major intrinsic protein (MIP) family, are responsible for the selective transport of water as well as other small molecules such as Si in the form of silicic acid, urea, and boric acid [8,9]. AQPs are not only present in plants but also in animals [10], fungi [11], and bacteria [12]. In the case of the prokaryotic system, only two AQPs, i.e., typical water transporter AqpZ and glycerol transporter GlpF, have been reported [13].

Microorganisms play a very important role in weathering soil minerals and making them soluble in a form readily available to plants. Exploration of such nutrient solubilizing bacteria can lead to a sustainable and eco-friendly approach [14]. The understanding of the molecular mechanism involved in the bacterial activity of nutrient solubilization is crucial for its efficient utilization. Whole-genome sequencing (WGS) of *Enterobacter* sp. LR6 aided the discovery of the enzyme involved in the solubilization of plant nutrients such as phosphate and silicate. The WGS of phosphate solubilizing bacteria *Pseudomonas lutea* performed by Y. Kwak et al. [15] provides a basic understanding of the molecular mechanism involved in the solubilization of insoluble phosphate by the action of phosphatase enzymes such as alkaline and acid phosphatases (AcPase). Identification of novel enzymes provides an opportunity for industrial applications of enzyme immobilization. Similarly, a molecular understanding of microbial activity will be helpful in understanding the entire biogeochemical cycle [16]. Very little is known in the case of microbial activity involved in silicate solubilization. A whole-genome sequence of any SSB is not yet available.

In the present study, we report the genome sequence of an SSB, *Enterobacter* sp. LR6, to identify aquaporins involved in Si transportation and plant-growth-promoting activity. Furthermore, comparative genome analysis of the *Enterobacter* sp. LR6 with closely related type strains was also performed to understand its evolutionary relatedness. The genes linked to arsenic resistance and stress responses such as acid, alkaline, salinity, and temperature were also explored to understand the survival strategy of the plant-growth-promoting *Enterobacter* in coping with extreme environmental conditions.

## 2. Materials and Methods

### 2.1. Identification and Phylogenetic Analysis of Bacterial Strain

Rhizospheric soil of a paddy field located at Mohali, Punjab (coordinates 30°40′03.3″ N 76°43′08.0″ E) was used to isolate SSB. The rice plants were uprooted and vigorously shaken to remove excess soil. Precisely 1 g of rhizospheric soil was dried at 30 °C for 12 h, serially diluted, and spread on the plate containing Luria–Bertani (LB) agar. The plates were incubated at 37 °C for 24 h. The isolates were then evaluated for silicate solubilization activity on Bunt and Rovira agar medium supplemented with 0.25% of Insoluble magnesium trisilicate [4]. The isolates were also checked for their ability to solubilize insoluble phosphate in NBRIP medium [17]. The purified isolate (10 µL) was spotted on the agar plate, followed by incubation at 30 °C for 2 days. The formation of a clear zone around the colony determines its ability to solubilize insoluble silicate and phosphate. The clear zone of the isolate was measured in terms of solubilization index (SI) determined as the ratio of total diameter (colony + holo zone) to the diameter of a colony [18].

The isolate positive for silicate solubilization was selected for DNA sequence-based characterization. The DNA was extracted from the overnight grown culture by using the Wizard bacterial genomic DNA purification kit based on the manufacturer’s guidelines. The 16s rRNA was amplified with the help of PCR using universal primers 27f (5′-AGAGTTTGATCCTGGCTCAG-3′) and 1492r (5′-TACGGYTACCTTGTTACGACTT-3′) supplied by Eurofins Genomics India and DNA as a template [19]. The quality and size of the amplified PCR product was analyzed on agarose and purified using a gel extraction kit. The purified product was sequenced with a BigDye^®^ Terminator v3.1 Cycle Sequencing Kit using the same universal primers 27f and 1492r [20]. The phylogenetic relationship of LR6 was established with closely related species using sequence information retrieved from the NCBI database (https://ncbi.nlm.nih.gov/, accessed on 5 May 2022). Similar 16s rRNA sequences having high coverage and percent identity were retrieved from the NCBI database and subsequently used to construct a phylogenetic tree in MEGA 11 using the maximum likelihood method with 1000 bootstraps [21].

### 2.2. Genome Sequencing and De Novo Assembly

Based on the Si solubilizing activity, phylogenetic distribution and potential multiple uses in the cropping system, *Enterobacter* sp. LR6 strain was selected for further extensive characterization through whole-genome sequencing. The DNA of the LR6 strain was isolated using a Promega Wizard^®^ bacterial genomic DNA purification kit. The quality of the isolated DNA was assessed using Nanodrop (Thermo Fisher Impulse Bioscience, Chandigarh). The DNA fragment library was prepared with 100 ng DNA using the Illumina Nextera DNA Flex Library Prep (Illumina DNA Prep; Cat. No. 20025519, 20025520, 20018704, and 20018705). The sequencing-ready libraries were prepared as instructed by the manufacturer. A Qubit instrument and a DNA HS kit (Invitrogen, Impulse Bioscience, Chandigarh) were used for the quantification of the libraries. To perform the multiplex sequencing, library concentrations were adjusted to 4 nM and pooled together. Pooled libraries were denatured and diluted at 300 pM and loaded in the library tube provided with the NovaSeq SP sequencing kits (300 cycles). The pooled libraries were further processed for paired-end (2 × 151 bp) sequencing on an Illumina NovaSeq 6000 instrument. The quality of the raw sequences was assessed using FastQC [22]. Then, the trimmomatic tool (v0.39) was used to process the raw reads based on quality along with adapter trimming [23]. The processed sequencing reads were assembled using an abyss-pe assembler with a k-mer size of 121 (after optimizing k-mer length from 25 to 127).

### 2.3. Whole-Genome-Based Taxonomic Analysis

The whole-genome taxonomic analysis of the strain LR6 was performed using a pairwise comparison of the genome with the genomes of other closely related type strains obtained from NCBI (https://www.ncbi.nlm.nih.gov/genome/, accessed on 6 May 2022). The type strains used for the analysis were *Enterobacter bugandensis* strain 247BMC, *E. hormaechei* subsp. xiangfangensis strain 10–17, *E. cancerogenus* strain LMG 2693, *E. cloacae* strain ATCC 13047, *E. asburiae* strain JCM6051, *Pantoea agglomerans* strain JCM1236, and *Cedecea lapagei* strain DSM 4587. The relatedness of strains was compared using digital DNA to DNA hybridization (dDDH) and average nucleotide identity (ANI) [24]. The dDDH values and confidence intervals were calculated at the default setting of the Genome-to-Genome Distance Calculator (GGDC) 3.0 (https://ggdc.dsmz.de/ggdc.php#, accessed on 12 May 2022). The OrthoANIu algorithm-based EZ BioCloud ANI Calculator tool [25] was used to estimate the ANI of the isolates using closely related genomes as reference. Orthologous Average Nucleotide Identity Tool (OAT) v 0.93.1 (Chunlab Inc., Seoul, Korea) [26] was used for the development of the heatmap and dDDH and ANI matrix among genomes. A phylogenetic tree based on the whole-genome sequence was constructed using the reference sequence alignment-based phylogeny builder (REALPHY) v1.13 by uploading the LR6 genome and all the reference genomes with default parameters [27].

### 2.4. Gene Prediction and Functional Annotation

The genome assembly was uploaded to Rapid Annotation using Subsystem Technology (RAST) v 2.0 server for rapid annotation. The service identified the protein-coding genes, assigned a function to identified genes, and also predicted the presence of rRNA and tRNA. The identification of rRNA and tRNA genes in RAST was performed by using “search_for_rnas” and tRNAscan-SE, respectively. The Transporter Automatic Annotation Pipeline (TransAAP) tool hosted at transportDB was used to identify the transporter families in the genome sequence of LR6. The CRISPR finder tool was used for the identification of clustered regularly interspaced short palindromic repeats (CRISPRs) in the genome [28]. For the identification of the secondary metabolite biosynthesis gene cluster Antibiotics and Secondary Metabolite Analysis Shell (antiSMASH) 5.0 was used [29]. The circular visualization of the *Enterobacter* sp. LR6 genome was developed using the proksee CG view server (https://proksee.ca/, accessed on 28 May 2022).

### 2.5. Comparative Genomic Analysis

For comparative genome-wide analysis, the genomic information of *Enterobacter* sp. LR6 and the closely related type strains *E. bugandensis* strain 247BMC, *E. hormaechei* subsp. xiangfangensis strain 10–17, *E. cancerogenus* strain LMG 2693, *E. cloacae* strain ATCC 13047, and *E. asburiae* strain JCM6051 were used. The MAUVE tool with default parameters was used for the genome comparison such as genome rearrangements and deletion between *Enterobacter* sp. LR6 and *E. hormaechei* subsp. xiangfangensis. The gene family clustering was determined with the help of Orthovenn2 using default parameters on the proteome sequences of *E. asburiae* JCM6051, *E. bugandensis* 247BMC, *E. cancerogenus* JCM6051, *E. cloacae* ATCC13047, and *E. hormaechei* subsp. xiangfangensis.

### 2.6. Identification, Characterization, and 3D Structure of Aquaporin

The identification of the aquaporin was performed with the help of BlastP search in BioEdit version 7.0.5.3 by creating a local database of the predicted protein sequence. The presence of the NPA motif and superfamily was determined by using the conserved domain database (CDD) at NCBI. The number of the trans-membrane domain was identified using TMHMM software. Isoelectric point and molecular weight of the proteins were determined using Expasy (web.expasy.org/compute_pi/, accessed on 28 May 2022). The subcellular localization was identified by using Psortb (https://www.psort.org/psortb/index.html, accessed on 28 May 2022) and Cello software (cello.life.nctu.edu.tw, accessed on 28th May 2022). The three-dimensional structure of the GlpF was predicted by using Alphafold (https://www.psort.org/psortb/index.html, accessed on 28 May 2022). CHAP: A Versatile Tool for the Structural and Functional Annotation of Ion Channel Pores, was used for the development of 3D structure using a Protein Data Bank (PDB) file generated from Alphafold. The pore morphology and position of hydrophobic and hydrophilic residues were also determined by using the CHAP tool [30].

### 2.7. Functional Validation of GlpF for Metalloid Transportation

For the functional validation of Si transporters, a yeast heterologous system was used. The vector pYES2 was digested with NotI and XmaI and cloned with GlpF. The resulting cloned vector and empty vector were then transformed in the yeast cell BY4741 (MATa his3Δ1 leu2Δ0 met15Δ0 ura3Δ0) using the PEG/LiAC transformation method [31]. The transformed *Saccharomyces cerevisiae* was grown overnight in 2 mL SD-URA glucose-supplemented media at 30 °C. The overnight grown culture was diluted to OD_600_ = 0.2 in a galactose-rich medium and incubated for 2 h at 30 °C. After 2 h of incubation, a dilution series of 10^−1^ to 10^−4^ was used for the yeast survivability assay. The ability of the transporters to uptake Si can be determined by using GeO_2_ as it generates germanic acid in the solution that was reported as an excellent tracer of silicic acid in the oocyte assay [32]. Precisely 10 μL of diluted culture was spotted on selection media containing arsenite As (III) and germanium di-oxide (GeO_2_) to determine its activity to transport As (III), and GeO_2_.

### 2.8. Effect of LR6 on Plant Growth

In order to investigate LR6’s ability to promote plant growth, soybean genotype JS335 was used. The soybean seeds were initially germinated for 7 days in soilrite and transferred to hydroponic assembly. All the containers were filled with the Hoagland nutrient medium (pH 5.5) to support the plant’s growth. A set of five replicates per treatment was transferred to the hydroponic assembly fitted with aeration pumps to provide adequate oxygen to the medium. After 20 days of treatment, three plants from the control and three from the SSB treatment were phenotyped for shoot/root length, fresh tissue, and dry tissue weight.

### 2.9. Data Availability

The 16s rRNA gene sequence and draft genome sequence of the *Enterobacter* sp. LR6 was submitted to NCBI with the accession numbers ON479461 and JAMZAU000000000, respectively. The version described in this manuscript is JAMZAU010000000.

## 3. Results and Discussion

### 3.1. Silicate-Solubilizing Activity of Enterobacter sp. LR6

Bacteria isolated from the rhizospheric soil were evaluated for the ability of silicate solubilization. Out of all the 63 isolates, only *Enterobacter* sp. LR6 was identified as SSB and phosphate solubilizing bacteria (PSB) based on its ability to solubilize by the formation of a visible hollow zone of dilution around the colony on Bunt and Rovira agar and NBRIP medium, respectively (Figure 1a,b). The SSI and PSI of the LR6 colonies were 2.1 ± 0.1 and 1.89 ± 0.1 mm, respectively. LR6 is a moderate silicone solubilizer and low phosphate solubilizer. According to L, Santi and D, Goenadi in 2017, bacteria having the Si solubility index 2.00 < SSI < 4.00 are considered as moderate silicate solubilizers [33]. Similarly, the bacteria showing phosphate solubilization index less than 2 are considered as low phosphate solubilizers [34]. Based on the solubilization ability, the isolate was then selected for further analysis.

### 3.2. Genome Structural Feature

The assembled genome of *Enterobacter* sp. LR6 has been deposited at GenBank with the accession number JAMZAU000000000 and comprises 5,226,212 bp, 55 contigs, and a mean GC content of 54.9% based on RAST annotation. A total of 5182 coding sequences and 98 non-coding RNA sequences were present in the genome (Figure 2). Out of the total RNAs, 86 were found to be tRNAs, and 12 were rRNAs. This number of rRNAs and tRNAs is typically a feature of soil bacteria and an indication of positive selection [35]. The presence of a high number of tRNA is a characteristic of soil bacteria which allows them to respond rapidly to changes in nutrient availability [36,37]. A total of three CRISPR elements were found in *Enterobacter* sp. LR6, similar to the number of CRISPRs present in its closest strain, *E. hormaechei* subsp. xiangfangensis.

### 3.3. Phylogenetic and Taxogenomic Analysis

To define the taxonomic classification of the *Enterobacter* sp. LR6, three main approaches were used: (i) multiple sequence alignment of highly conserved 16S rRNA sequences, (ii) calculation of dDDH and ANI, and (iii) whole-genome phylogenetic analysis using closely related bacterial species. The phylogenetic tree constructed based on the 16S rRNA showed that *Enterobacter* sp. LR6 is closely related to NR_118568.1 *E. cloacae* and NR_126208.1 *E. hormaechei* subsp. xiangfangensis with a percent similarity of 97.16% and 96.74%, respectively (Figure 1b). The genomic similarity was also assessed based on the ANI score, which also indicates a close relationship of LR6 with the *E. hormaechei* subsp. xiangfangensis. Since it showed an ANI value of 97.12, which is higher than 95–96%, a speciation threshold is recommended for a strain belonging to the same species [38]. The dDDH analysis of LR6 with the *E. hormaechei* subsp. xiangfangensis showed a dDDH score of 72.6%. The dendrogram developed based on the ANI and dDDH values showed that the *Enterobacter* sp. LR6 was found to be closer to *E. hormaechei* subsp. xiangfangensis with an ANI value of 97.12% and GGDC distance of 0.03 (Figure 2c,d). The whole-genome phylogenetic tree constructed using the REALPHY tool also confirms the closeness of LR6 to the *E. hormaechei* subsp. xiangfangensis as observed with the ANI and dDDH dendrogram.

Comparative sequence analysis based on 16s RNA is one of the most important methods for the identification of bacteria and phylogeny construction to determine the relationship with closely related type strains. The comparison of prokaryotes was previously only based on their phenotypic similarities, but at present, effective and fast genome comparison enables more accurate taxonomical positioning. Here, the genome sequence comparison of LR6 among the closely related type strains was greatly influenced by advanced genetic methods, including ANI and dDDH. The exponential development in the field of sequencing technologies and increasing numbers of genome sequences make the process easy for whole-genome distance determination.

### 3.4. Comparative Genome Analysis

The genome comparison between the closely related type strains of LR6 shows the difference between the genome sizes ranging from 4.6 to 5.3 Mb and GC content from 54.81% to 56.02%. The comparison of genome sequences by using Orthovenn2 identified 4860 clusters which include 3069 core genome orthologs. The unique proteins in the various sequences vary from 92 in *Enterobacter* sp. LR6, 3 in *E. hormaechei* subsp. xiangfangensis and *E. bugandensis* 247BMC, 4 in *E. cloacae* ATCC13047, 6 in *E. cancerogenus* JCM6051, to 31 in *E. asburiae* JCM6051 (Figure 3a). Further comparison of LR6 with closely related species *E. hormaechei* subsp. xiangfangensis revealed the sharing of a total of 3816 core genome orthologs, and the specific gene clusters were 150 and 25, respectively (Figure 3c). The genome-wide dynamics of *Enterobacter* sp. LR6 and *E. hormaechei* subsp. xiangfangensis was analyzed using the Mauve comparative genome analysis tool. The analysis of these genomes showed 30 LCBs with a minimum weight of 341 and a maximum of 2,84,730. The minimum LCB weight could be due to the comparison of the LR6 draft genome with the complete *E. hormaechei* subsp. xiangfangensis genome (Figure 3e).

The comparison of the genome with the closely related type strains showed that the draft genome of *Enterobacter* sp. LR6 is larger (5.2 Mb) than the *E. hormaechei* subsp. xiangfangensis 4.6 Mb, *E. bugandensis* 247BMC 4.6 Mb, *E. cloacae* ATCC13047 4.8 Mb, *E. cancerogenus* JCM6051 4.7 Mb, and *E. asburiae* JCM6051 4.8 Mb; this difference may be due to the presence of few fragments of plasmid in the genome sequence. *Enterobacter* sp. LR6 has a G + C content (54.94%) lower than all related type strains mentioned previously (55.28, 56.02, 55.03, 55.77, and 55.47%, respectively).

### 3.5. Functional Annotation of the Genome

As per the RAST annotation, 752 genes were identified for biomolecule metabolism, which includes 338 genes for carbohydrate metabolism, 227 for protein metabolism, 48 for fatty acid and lipid metabolism, and 139 for nucleic acid (56 RNA and 83 DNA) metabolism. The genome of LR6 also contains 149 genes for the metabolism of the different important materials, including 17 genes for potassium metabolism, 35 for nitrogen metabolism, 25 for sulfur metabolism, 29 for phosphorus metabolism, and 43 for iron acquisition and metabolism. The genes associated with the different functions such as virulence, disease, defense, respiration, sporulation, and stress response were also identified in the genome (Appendix A). Further, a total of 12 phages and 6 prophage regions were identified with the help of the PHASTER tool (Appendix A). In addition, the *Enterobacter* sp. LR6 also contains four clustered regularly interspaced short palindromic repeats (CRISPRs). CRISPRs are likely to provide acquired resistance to bacteriophages. A total of 723 genes for membrane transporters were identified by the Transporter Automatic Annotation Pipeline (TransAAP). A total of 279 ABC transporters, 9 F-ATPase, 9 P-ATPase, 88 MFS, 61 SSPTS, 2 MIPs, and several other transporters predicted to be involved in the transportation of elements such as As, zinc, iron, and chromium, were identified (Appendix A).

### 3.6. Prediction of Gene Clusters Encoding Secondary Metabolite Stress Tolerance and Colonization-Related Protein

The genome of *Enterobacter* sp. LR6 contains secondary metabolite gene clusters such as NRPS, thiopeptide, redox cofactor, siderophore, and arylpolyene identified using the antiSMASH 5.0 tool. The total fraction of genes in these clusters includes 38 genes in the non-ribosomal peptide synthetase (NRPS) cluster, 21 in the thiopeptide, 20 in the redox cofactor, 10 in the siderophore and 46 in the arylpolyene cluster. The presence of this useful biosynthetic gene cluster in *Enterobacter* sp. LR6 helps the bacteria to adapt to the extreme environment. The *Enterobacter* sp. LR6 can survive in conditions of up to 45 °C, 7% salinity, and pH 9 (Appendix A). The members of Enterobacteriaceae have been identified as an opportunistic pathogens with multidrug resistance [39]; thus, risk analysis is necessary. The risk of potential pathogenicity of LR6 to human or other animals was evaluated based on β-hemolysis activity on sheep blood agar plate (Hi-media). The β-hemolysis activity was absent in the case of LR6 as it did not form any hemolytic zone around the colony (Appendix A). It is rare to find bacteria that are nonpathogenic and beta-hemolytic at the same time [40]. The siderophore produced by the bacteria limits the growth of soil-borne phytopathogens [41]. Aryl polyenes can protect bacteria from reactive oxygen species [42], the redox cofactor cluster helps in the transformation of different complex organic compounds [43], and the thiopeptide cluster helps the bacteria and plants by inhibiting the growth of gram-positive bacteria in the rhizospheric region [44]. The biosynthetic pathways of the secondary metabolites from soil microorganisms are currently subject to intense efforts due to their high potential in synthetic biochemistry and agriculture.

### 3.7. Genetic Potential of Enterobacter sp. LR6 for Survival in Plant Rhizosphere

The process of rhizospheric colonization of any plant-growth-promoting bacteria to the host plant can be divided into main three processes, (i) movement towards host plants through chemotaxis, (ii) adhesion to the host surface, and (iii) stress adaptation around the root. The genome annotation and comparative genome analysis revealed that *Enterobacter* sp. LR6 seems to be very adaptive in the plant rhizosphere.

#### 3.7.1. Movements towards Rhizosphere: Motility/Chemotaxis

The initial step for the colonization involves the movement of bacteria toward the rhizosphere. Motility is an important characteristic of soil bacteria. Although the bacteria in soil can move passively with the water flux, the induction of flagellar activity by plant-released compounds influences the active movement. The genome of LR6 contains flagellar synthesis operons such as *flgNMABCDEFGHIJKL, flhEAB, cheZYBR, cheWA, motBA, flhCD, fliCDSTEFGHJKLMNOPQR,* and *fliEFHIJKLMNOPQR* which follow a similar trend to *Salmonella enterica* subsp. *Enterica typhi* [45]. Additionally, several genes and proteins responsible for the chemotaxis activity such as putative chemotaxis protein (*cheA*), methyl-accepting chemotaxis sensory transducer (*mcp*), methyl-accepting chemotaxis citrate transducer (*tcp*), methyl-accepting chemotaxis protein I (*tsr1, tsr2*), methyl-accepting chemotaxis protein II (*tarA*), methyl-accepting chemotaxis protein III (*trg*), methyl-accepting chemotaxis protein IV (*tap*), and chemotaxis protein (*cheZ, cheR, cheV*) were identified.

#### 3.7.2. Hemagglutinin

The colonization of the bacteria in the rhizospheric region depends on the variety of cell surface-associated factors that are responsible for the adhesion of bacteria to the host surface. The protein that plays an important role in adhesion is filamentous hemagglutinin-like adhesin [46]. The genome of LR6 encodes a putative hemagglutinin-related protein responsible for adhesion.

#### 3.7.3. Counteracting the Plant’s Defense Mechanism

The plant may use a variety of defense mechanisms against the soil-borne bacteria, including the production of reactive oxygen species (ROS) such as hydrogen peroxide, superoxides, hydroperoxyl radicals, phytoalexins, and nitric oxide. To colonize the rhizospheric region of plants, LR6 has to survive in the harsh oxidative environment around the rhizosphere of plants. The genome of LR6 possesses three superoxide dismutases, three catalases, alkyl hydroperoxide reductase, hypoperoxide reductase, and thiol peroxidase that can protect LR6 from the oxidative stress created by the plant. LR6 is also able to detoxify free radical nitric oxide and overcome the phytoalexins for successful colonization of the host rhizosphere (Table 1).

#### 3.7.4. Stress Management

The *Enterobacter* sp. LR6 genome carries several genes reported to be involved in several stress adaptations including heavy metal stress, temperature stress, acid stress, and alkaline stress. In order to combat heavy metal stress, it carries the genes for type ATPase CopA regulated by CueR, multiple copper oxidase CueO, and an operon that codes for the copper resistance proteins CopC and CopD. Other heavy metal resistance genes present in the genome provide resistance to arsenic, such as ArsA and ArsB. The AsrA protein is an ATPase that associates with AsrB for the formation of an arsenite efflux pump, energized by ATP hydrolysis. Osmolarity sensory histidine kinase EnvZ and OmpR (transcriptional response regulator) are the stress sensor proteins that impart the osmotic stress tolerance to LR6 [66]. Additionally, the genes responsible for acid stress, alkaline stress, and several other heavy metal stress genes were identified in the genome (Table 1).

### 3.8. Plant-Growth-Promoting Activity

Plant-growth-promoting activity of the *Enterobacter* sp. LR6 is related to its phosphate solubilization and mineralization property. The enzyme responsible for phosphate mineralization, i.e., alkaline phosphatase (*phoA*), was found in its genome. The other genes that allow the bacteria to uptake and solubilize were also found in the LR6 genome including phosphate starvation-inducible protein (*phoH*), phosphate transport system regulatory protein (*PhoU*), pi-specific transporter (*pstA, pstB,* and *pstC*), phosphate regulon sensor protein (*PhoR*), phosphate regulon transcriptional regulatory protein (*PhoB*), and glycerol-3-phosphate ABC transporter (*UgpA*). Along with the phosphate solubilization gene, LR6 contains a gene that is responsible for the production of enzyme that may have silicase activity. The silicase enzymes belong to the carbonic anhydrase (CA) family (EC 4.2.1.1). This enzyme is responsible for the hydration of SiO_2_, like the hydration reaction of CO_2_ by CAs [67,68]. Considering the benefits of Si in plants, the increased concentration of soluble Si in soil can promote their overall growth.

The genome of LR6 also contains the gene responsible for the production of pyruvate decarboxylase (EC 4.1.1.1) (Appendix A), a key enzyme in the biosynthetic pathway for the production of indole-3-acetic acid. It catalyzes the indole-3-acetaldehyde and carbon dioxide from indole-3-pyruvic acid [69]. The *Enterobacter* sp. LR6 is also able to produce a group of heterogeneous molecules called siderophores that can sequestrate iron. These siderophores have an antagonistic effect on the other microorganisms by depriving them of iron and ultimately protecting the plants from pathogenic infections. The genome of LR6 contains genes for the synthesis of bacterioferritin (*bfr*), siderophore enterobactin (*EntB, EntS*), ferric and ferric-related siderophore receptors (*TonB*), ferrous iron uptake systems (FeoAB, EfeUOB), two ABC transporters (*sitABCD* and *fepCGDB*) involved in iron uptake, and outer membrane siderophore receptor (*IroN*).

A study by A Ramesh, SK Sharma, MP Sharma, N Yadav and OP Joshi [70] emphasized the role of *E. cloacae* in soybean plant growth promotion. Hogland medium inoculated with LR6 significantly improved plant growth compared to the control (Figure 4). However, the average height of the control shoot (i.e., 81.7 ± 1.7 cm) and root (42.3 ± 1.7 cm) was higher than the SSB-treated shoot and root, i.e., 78.3 ± 2.8 and 34.8 ± 1.5 cm, respectively, but fresh and dry weight of SSB-treated plant (i.e., 66.8 ± 1.7 and 19.7 ± 0.5 g, respectively) was much higher than the control (i.e., 42.9 ± 1.2 and 16.29 ± 0.7 g, respectively) (Figure 4e–h), which indicates the plant growth promotion potential of LR6.

### 3.9. Identification, Characterization, and Functional Validation of GlpF

Only two AQPs (one AQPZ and one GlpF) were identified in the LR6 genome based on sequence homology with previously identified AQPs. The CDD search confirms that both the identified genes belong to the MIP superfamily and contain the NPA (asparagine–proline–alanine) signature motif. Based on the TMHMM search, both the AQPs contain six transmembrane helices. Both the AQPs are localized on the inner membrane and cytoplasmic membrane based on the Cello and pSortb search, respectively. The AQPZ is a classical AQP and is responsible for the transportation of water [71], whereas GlpF is responsible for the transportation of water, glycerol, and metalloids [72,73]. The phylogenetic tree of aligned GlpF and AqpZ from different bacteria was constructed using the MEGA11 program via the maximum likelihood method (MLM) (Appendix A). The identified GlpF in Enterobacter showed 89.68% identity with well-characterized *E. coli* GlpF (Figure 5d). The structure for LR6 GlpF was generated using Alphafold, and the top-scoring model was used for further pore feature characterization (Figure 5a). In agreement with other GlpF, the Ar/R filter consisted of WGFR and formed the size constriction of the pore (Figure 5b). In several studies, the involvement of GlpF has been reported in the uptake of several metalloids such as As (III) and antimony Sb (III) [73]. To validate the involvement of GlpF in the uptake of silicic acid, the GlpF was heterologously expressed in yeast. As a positive control, we used Nodulin 26-like Intrinsic Protein (OsNIP2;1), a Si transporter in rice. The yeast transformants were plated on the medium containing various concentrations of As (III), GeO_2_, and boric acid. To study the transport activity of the GlpF in the yeast heterologous expression system, the minimum lethal concentration of As (III) and GeO_2_ for the wild-type strain was optimized. The growth of yeast expressing OsNIP2;1 and GlpF was inhibited on the medium containing 1.25 mM As (III) and 5 mM GeO_2_, whereas the control yeast transformed with the empty vector grew at a higher density at the same concentration (Figure 5c).

The structural comparison between the *E. coli* and LR6 GlpF has shown that both the GlpF crystallize as a symmetric arrangement of four channels. The monomer of both the GlpF consists of six transmembrane helices (TMHs) and two half-spanning helices. The GlpF of both the microorganisms shows a high degree of similarity in terms of selectivity filters (SFs) such as the presence of W48, G191, F200, and R206 at the same position. The radius of the Ar/R filter is similar at about 1.9 in the case of LR6 GlpF and around 1.85 in *E. coli* GlpF. The GlpF not only facilitates the transport of water, but it can also transport polyalcohols such as glycerol and metalloids such as As and Sb [73]. A similar trend has been followed by the LR6 GlpF for the transport of metalloids such as As and Ge.

## 4. Conclusions

In this study, we isolated the *Enterobacter* sp. LR6, an SSB from the rhizospheric soil of rice. LR6 was found to solubilize silica optimally in Bunt and Rovira media. In terms of its SSI, LR6 can be considered as a moderate silicone solubilizer. Apart from silicate solubilization potential, the whole-genome sequencing of LR6 performed here highlighted genes important for several plant-growth-promoting traits such as phosphate-solubilizing properties and siderophore production, stress adaptation against oxidative stress, acid stress, alkaline stress, temperature stress, osmotic stress, and heavy metal stress from arsenic, copper, chromate, cobalt, and zinc. This study emphasized mainly the presence of genes in SSB potentially responsible for bacterial movement towards the rhizosphere, its colonization, plant growth promotion, and Si uptake. The bacterial aquaporin GlpF is involved in the uptake of As (III) and GeO_2_. The uptake of GeO_2_ by the LR6 GlpF confirmed by the yeast assay also indicate the possibility of Si transport through GlpF. LR6 was assessed for potential pathogenicity based on its β-hemolysis activity. The β-hemolytic activity in LR6 was absent as there was no hemolytic zone around the colony. In spite of this, Enterobacteriaceae members are known to be opportunistic pathogens; hence, further risk analysis is required. In addition, LR6 was found to tolerate high temperature, salinity, and higher pH. Therefore, LR6 can provide a sustainable solution to improve crop yield. In addition, the comprehensive genome sequencing information of the *Enterobacter* sp. LR6 provided a landscape highlighting genes involved in silicate solubilization, stress tolerance, and growth-promoting activity. The information provided here will serve as a basis for understanding molecular mechanisms involved in silicate-solubilizing activity of LR6. Specifically, the LR6 genomic sequence will serve as valuable resource for gene identification. Similarly, the information will be helpful to exploring SSB as a biofertilizer.

## Figures and Tables

**Figure 1 cells-11-03622-f001:**
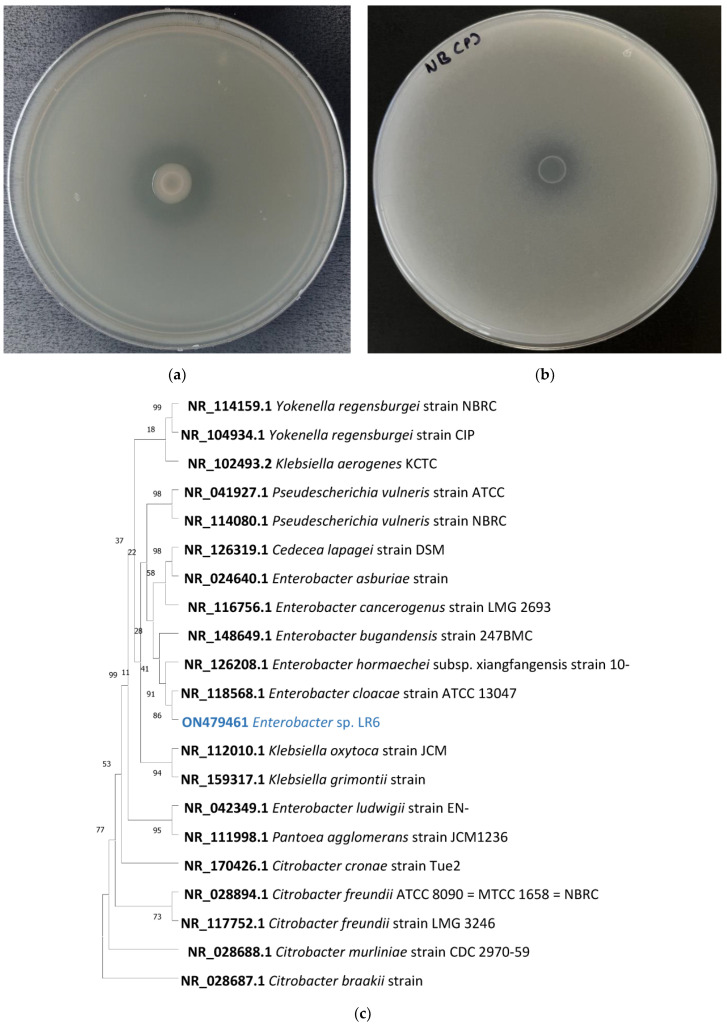
*Enterobacter* sp. LR6 showing silicate solubilization on Bunt and Rovira silicate medium (**a**), phosphate solubilization on NBRIP medium (**b**), 16s rRNA gene-sequence-based phylogenetic tree constructed using the maximum likelihood method. Based on the 16s rRNA search, the closely related type strain was aligned using ClustalW, and the phylogenetic tree was made using the 1000-bootstrap method. NR_028687.1 *Citrobacter braakii* strain 167 was selected as the outgroup. The values shown on the tree indicate the bootstrap percentage (**c**).

**Figure 2 cells-11-03622-f002:**
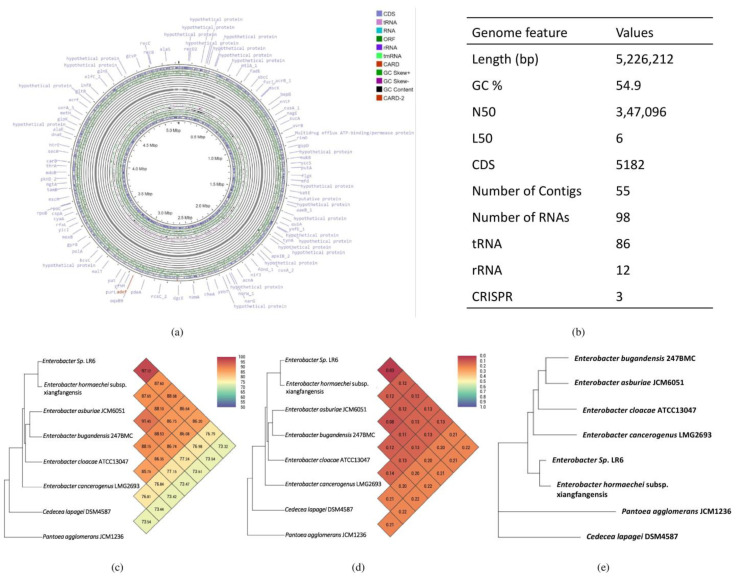
Graphical circular map of *Enterobacter* sp. *LR6*. The green color and gray color represent the open reading frame and contigs, respectively. The outer circle in orange and blue color shows predicted antibiotic resistance and protein-coding sequences (**a**), the table showing its general genome feature (**b**), heat map and phylogenetic tree based on (**c**) ANI, (**d**) GGDC distance, and (**e**) whole-genome sequence of *Enterobacter* sp. with its closely related taxa.

**Figure 3 cells-11-03622-f003:**
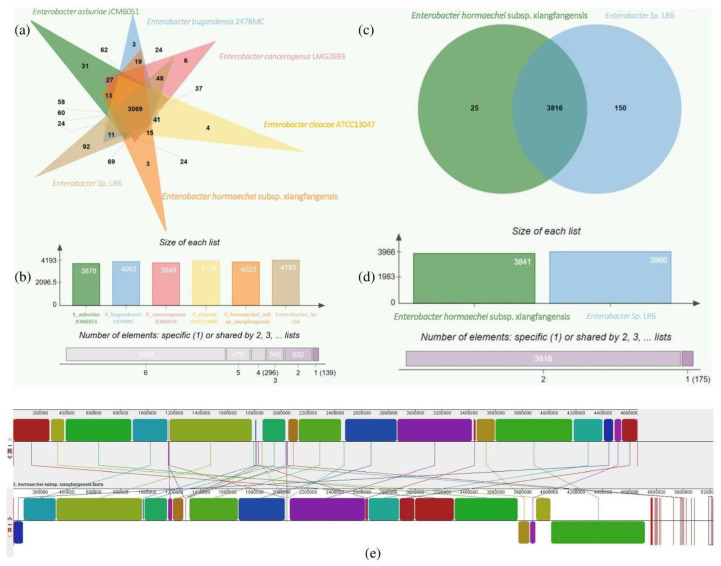
Venn diagram representing the distribution of unique and shared proteins among *Enterobacter* sp. LR6 and closely related genomes (**a**); the bar plot below the Venn diagram shows the number of proteins found in each species (**b**). Distribution of *Enterobacter* sp. LR6 proteins with the closest strain *E. hormaechei* subsp. Xiangfangensis genome (**c**); the bar plot shows the number of proteins found in *Enterobacter* sp. LR6 and *E. hormaechei* subsp. Xiangfangensis (**d**). Genome alignment of *Enterobacter* sp. LR6 and *E. hormaechei* subsp. Xiangfangensis genome constructed using MAUVE showing the genome-wide variation. The local collinear blocks (LCBs) in different colors represent conserved segment, white areas within LCBs represent regions with low similarity, and the scale represents the coordinates of each genome. The LCBs present above and below the central black horizontal line are the forward and reverse orientation, respectively. The rearrangement between the two genomes is represented by the colored lines (**e**).

**Figure 4 cells-11-03622-f004:**
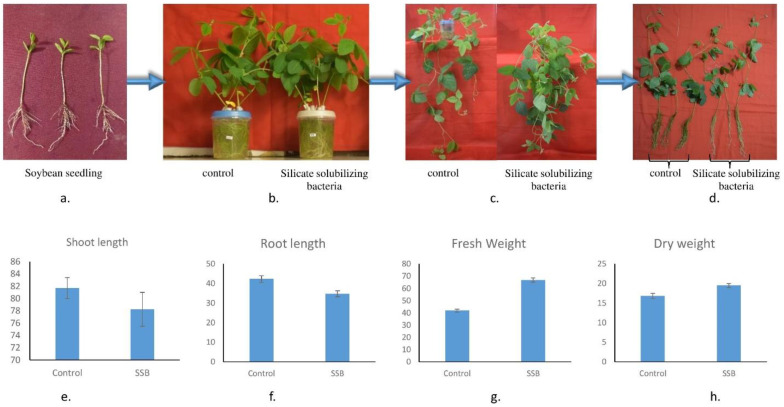
Comparison of control and LR6-treated plant. The 7-day seedlings (**a**) were transferred to a customized hydroponic system (**b**), and the difference was visualized after 30 days (**c**,**d**). Comparison of root, shoot length and fresh weight and dry weight of control vs. silicate-solubilizing bacteria-treated plants (**e**–**h**).

**Figure 5 cells-11-03622-f005:**
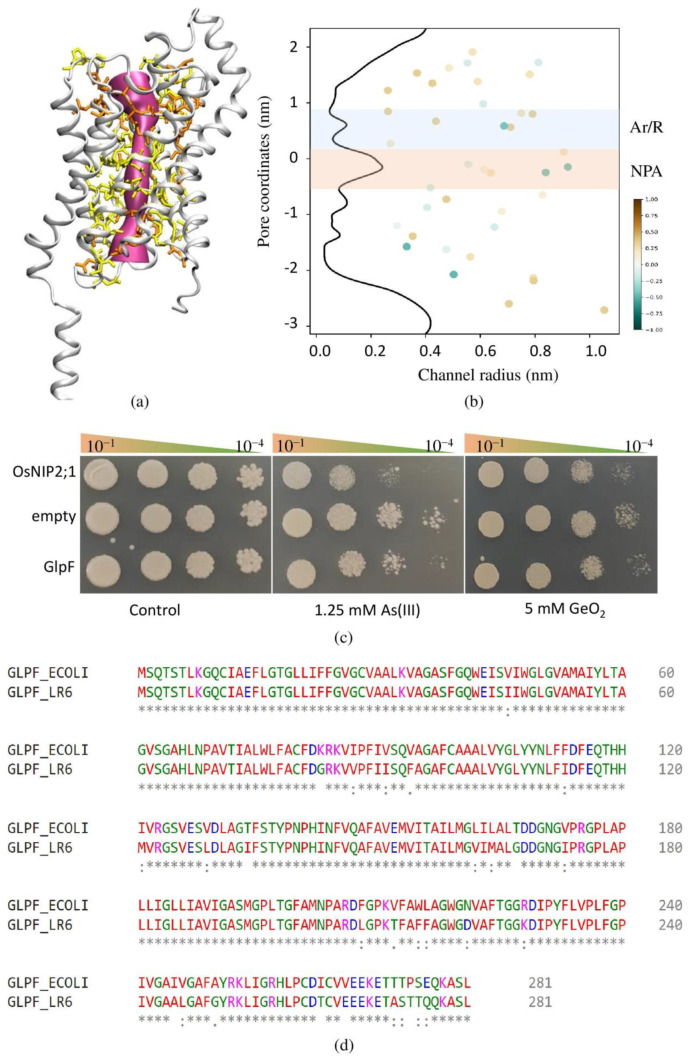
Three-dimensional structure of Enterobacter GlpF. Pore-facing residues are represented in orange, while the pore-lining residues are in yellow (**a**). The pore radius for predicted Enterobacter GlpF plotted on z coordinates of protein. The position of hydrophilic and hydrophobic residues was spotted at the corresponding position. The representation of hydrophilic to hydrophobic residue is shown on green to brown scale using CHAP tool. The region shaded in light blue and orange color indicate the location of Ar/R residues and NPA motifs (**b**). Growth of yeast cell expressing aquaporin homolog of OsNIP2;1, GlpF transformed with pYES2 and empty vector. A total of 10 µL of diluted yeast transformant (10^−1^ to 10^−4^ dilution) was spotted on media containing various concentration of As (III) and GeO2. The growth was recorded after 4 days at 30 °C (**c**). Sequence alignment of LR6 GlpF with *E. coli* GlpF (**d**).

**Table 1 cells-11-03622-t001:** The genes predicted to be involved in different stress adaptation mechanisms.

Stress Type		Protein Type	Genes/Operon/Protein	Location	Function	References
Oxidative stress		Superoxide dismutase	SodA (Mn superoxide dismutase); SodB (Fe superoxide dismutase); and SodC (Cu/Zn superoxide dismutase)	sequence01_2888396_2887776, sequence01_1115542_1114961, sequence01_1123281_1123799	Provides superoxide resistance	[47]
		Catalase	KatE, manganese catalase, and KatG	sequence01_4550870_4549998, sequence01_3881760_3883940	Protection against the H_2_O_2_	[48]
		Hydroperoxide reductases	Organic hydroperoxide resistance protein, organic hydroperoxide resistance transcriptional regulator	sequence01_3509831_3509403, sequence01_3510368_3509916	Detoxification of organic hydroperoxide	[49]
		Alkyl hydroperoxide reductase	Alkyl hydroperoxide reductase protein F (ahpF), alkyl hydroperoxide reductase protein C (ahpC), alkyl hydroperoxide reductase subunit C-like protein	sequence01_419756_421321, sequence01_419004_419567, sequence01_178141_177539	Oxidative stress defense	[50]
		Thiol peroxidases	Thiol peroxidase, Tpx-type, thiol peroxidase, Bcp-type	sequence01_1694758_1695264, sequence01_2514125_2514595	Reduces t-butyl hydroperoxide, H_2_O_2_, and cumene hydroperoxide	[51]
		Nitric oxide dioxygenase	Flavohemoglobin/nitric oxide dioxygenase, nitric oxide reductase FlRd-NAD(+) reductase, anaerobic nitric oxide reductase flavorubredoxin	sequence01_2597555_2598745, sequence01_4699018_4697885, sequence01_4700463_4699015	Detoxifies free radical nitric oxide	[52]
		Nitrate reductase	Anaerobic nitric oxide reductase transcription regulator NorR	sequence01_4700651_4702165	Required for the expression of anaerobic nitric oxide (NO) reductase	[53]
		RND transporters	AcrAB	sequence01_266663_267316	Export of pytoalexins	[54]
Heavy metal resistance	Arsenic	Arsenate reductase, arsenic resistance protein, arsenite/antimonite:H+ antiporter, arsenical pump	ArsB, ArsA	sequence01_673660_672371, sequence01_4740871_4739582, sequence31_15724_17007, sequence01_4734229_4735986, sequence01_4742670_4740919	Provides resistance to arsenic	[55]
	copper	Copper resistance protein, copper resistance protein	CopD, Cop O, CopC, CopB, CueR	sequence01_1510825_1509896, sequence01_1931658_1930789, sequence01_1511210_1510830, sequence01_1932031_1931660, sequence01_1512146_1511250, sequence01_290196_290606	Provides resistance to copper	[56]
	chromate	Chromate reductase, transport and resistance protein	ChrA, ChrB	sequence01_4722045_4720666, sequence01_4722976_4721999	Provides resistance to chromate	[57]
	Cobalt, zinc and cadmium	Zinc transporter	ZitB, RcnR-like protein, RcnA, RcnB	sequence01_543130_542192, sequence01_1425411_1425686, sequence01_254854_254039, sequence01_4727409_4728539, sequence01_2243421_2243753	Cobalt–zinc–cadmium resistance	[58,59]
Temperature stress		RNA polymerase sigma factor	rpoE	sequence01_1576112_1575555, sequence01_2635482_2634907	Regulates the degQ and support growth at low and high temperature	[60]
		Stress sensor	degQ	sequence01_4075345_4073978	Enables bacteria to grow at high temperature	[60]
		Csp family	CspB, CspC, CspD, CspE	sequence01_1202383_1202165, sequence01_1668932_1668744, sequence01_1913740_1913531, sequence01_730174_729953, sequence01_441915_442124	Protects the bacteria during rapid downshift of temperature	[61]
		Hsp family	HspQ, HslJ, GrpE	sequence01_824127_823810, sequence01_1671988_1672425, sequence01_3834576_3835169	Involved in protein folding and refolding, expressed in high temperature	[62]
Acid stress		Acid stress protein	IbaG	sequence01_4123024_4123278	Changes mRNA expression pattern to provide acid resistance	[63]
		F0F1-ATPase transporter	F0F1-ATPase	sequence01_2972737_2973117, sequence01_2973126_2973941, sequence01_2974764_2975297, sequence01_2975310_2976851, sequence01_2976903_2977766, sequence01_2977798_2979180, sequence01_2979201_2979620	Induces acid tolerance response	[64]
		Arginine deiminase, ornithine carbamoyltransferase	arcA, arcB	sequence01_3548646_3547930, sequence01_4107667_4110000	Leads to production of alkaline molecules, such as ammonia to maintain pH.	[64]
Alkaline stress		Membrane-bound Na^+^/H^+^ antiporter	nhaA, nhaB, nhaP2	sequence01_3565391_3566566, sequence01_1886487_1888025, sequence01_1879396_1881129	A membrane-bound Na^+^/H^+^ antiporter system adapts the bacteria to alkaline stress	[65]
Osmotic stress		Osmolarity sensory histidine kinase, transcriptional response regulator, OmpR family	EnvZ/OmpR two-component system	sequence01_2738070_2736724, sequence01_2738786_2738067, sequence01_685891_686598, sequence01_1485204_1485944	Mediates osmotic stress response in a number of Gram-negative bacteria	[66]

## Data Availability

The 16s rRNA gene sequence and draft genome sequence of the *Enterobacter* sp. LR6 were submitted to NCBI with the accession numbers ON479461 and JAMZAU000000000, respectively.

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
