# Peer review of "Genomic Landscape Highlights Molecular Mechanisms Involved in Silicate Solubilization, Stress Tolerance, and Potential Growth-Promoting Activity of Bacterium Enterobacter sp. LR6"

_cells, 2022, doi:10.3390/cells11223622_

Round 1
Reviewer 1 Report (Previous Reviewer 1)
While the authors addressed the reviewer comments and offered additional experimental testing, there are still a few points that need to be clarified prior to publication:
Line 330-332 (pg. 18): Testing for risk does not just mean whether or not the strain has hemolytic activity. Many pathogens, just as Yersinia pestis, are non-hemolytic and pathogenicity arise from other modes of action. In the literature, based on a large screen, there were no identified hemolytic stains of Enterobacter spp (out of 53 tested) and only 32.9% of the clinically isolated E. coli were hemolytic. (Roland, F. P. 1977. Interaction of blood with Enterobacteriaceae: Hemolysis, Hemagglutination, Fibrinolysis. American Journal of Clinical Pathology 67(3): 260-263.). The clinically significant Enterobacter spp., E. cloacae and E. aerogenes, are also non-hemolytic. It is not suggested that the authors carry out a risk-assessment of the isolate, simply that with the growth conditions this particular isolate can survive in suggests its potential as an opportunistic pathogen.
Line 401-402 (pg. 20) …contains a gene that is responsible for the production of enzyme having silicase activity.’
Did the authors test for this activity? Carbonic anhydrase enzymes are a large family of genes found in many different cellular organisms. This may have putative enzymatic activity, but without proper testing, it cannot be definitively referred to as a silicase enzyme. Please consider revision to indicate that the gene has the potential to be a silicase enzyme.
Line 418:420 (pg. 20): This needs to be included in the Materials and Methods on the details of how the culture was grown and inoculated into the nutrient solution, how the solution was aerated, pH, etc. The authors stated that the growth of the plant was ‘significantly improved’ but no data analysis on growth are provided. Significantly how? More leaves, larger leaf area, longer shoots? How was this determined (i.e. ANOVA analysis) and how many plants were analyzed (i.e. n=?)?
Figure 4 (pg. 21): It is not clear which treatment in ‘C’ was the control and which was inoculated with LR6. Also, there is no data on how the growth was evaluated. One plant looks like it has longer internodes and the other possibly more leaf area. What is the desired trait for this genetic line of soybean? And why was this carried out in hydroponics as opposed to soil trials? There is no negative control for a bacterial isolate that is not considered a PGR.
Also, why is this figure put together with the pathogenicity test and GlpF alignment? These are three different data sets that should have their separate figures. The SBA could probably be moved to a supplemental file. Why isn’t the MSA included in Figure 5 with the other data on characterization of GlpF?
Inclusion of the materials and methods as well as a more precise data analysis of the growth characteristics in soybean should be included prior to acceptance for publication.
Author Response
Please see the attachment

Reviewer 2 Report (Previous Reviewer 2)
L2. Title overstates the findings. Should be “potential” or “possible” mechanism.
L4 L17. Italic “sp.” or not? Thoroughly check the manuscript.
L21. Should be “identify potential genes involved….”, as without experimental confirmation evidence.
L22. Carefully using “also”. Here without comparison.
L15-31. The abstract should be a total of about 200 words maximum.
L63. Pay attention to reference style. Three authors. “Kwak et al” maybe the right way.
L79. Upper case “analysis” and “strain” or not? Please use the same form.
L80. N or S and W or E for the coordinate.
L94. It should be “’”, not “'”. And without kit information such as producer, city, country. Thoroughly check the manuscript.
L95-96. Missing information for primer synthesis producer.
L105. Italic “de novo”.
L114. Why upper case “Library”?
L118. Missing reference. Thoroughly check the manuscript.
L152. Should using the abbreviation form of Latin name.
L175. Italic enzyme names.
L177. First appearance of S. cerevisiae, thus using full name.
L179. Adding “nm” is uncommon.
L180. Missing “.” after assay.
L183, 184. Make sure “(AsIII)” and “As (III)”.
Methods for LR6 supplementation improvement growth of soybean plants grown are missing.
L190, L191. “2” and “3.1”. Such kind of mistake is unacceptable.
L191. The subtitle is confusion. Rewrite.
L194. “PSB” without full name.
L197. Pay attention to reference style.
L213. Should be “5,182”. Thoroughly check the manuscript.
L237. Missing thousands separator in Fig 2b number. The bar of Fig 2a is too close to Fig 1b, suggesting move them close to Fig 1a. Latin name in Fig 2c and 2d should be full name and italic.
L273. Latin name in Fig 3a should be full name and italic. Why italic the labels, such as “Size of each list”? Veen panels should be sequentially marked individually. The current form, Fig 3a contains multiple panels, which should be marked with “a, b, c, d” and so on. In Fig 3c, ruler without unit, “bp” should be added.
L369. As LR6 was isolated from rice, the plant growth-promoting activity should also be tested on rice.
Author Response
Please see the attachment

Reviewer 3 Report (New Reviewer)
This manuscript presents data that allow us to link a whole complex of phenotypic traits of the Enterobacter sp. LR6 with its genomic characteristics. In general, the manuscript presents important data that is of interest to a wide audience. The authors touch upon a wide range of problems associated with plant-microbial interactions that promote plant growth and protect plants from biotic and abiotic stresses. The important shortcomings of the work are the following: 1) overload of the text with details behind which the main message can be lost; 2) the data shown in Table 1 and in lines 302- 316 and 322-339 is difficult to interpret as there is no comparison with other genomes. For example, one could take a related microorganism that does not show plant-microbial effects. In this case, the presence of genes that control adaptations to stress in the LR6, and their absence in the other, could give a clearer meaning to the data presented (However, I do not insist on this); 3) data on plant growth promotion performed on one plant without measurements cannot be considered reliable.
In general, the article is quite interesting, but I don't know what to do with plant growth promotion. If the authors do not have the opportunity to set up this experiment in sufficient replications and with adequate measurements, then it may be better to remove this part. The authors will have enough material for a good article.
Particular notes:
46-47 It is necessary to briefly explain by what mechanisms complexes with the organic hydroxy compounds, proteins, carbohydrates and suger derivatives can improve the overall growth of plants;
98-99 It is necessary to briefly explain for non-specialists how insoluble magnesium trisilicate can simulate the interaction with the much more inert (probably) silicon oxide;
139-143 What was the criterion for choosing a reference group of genomes? Is this a top list by similarity?
229-230 What do you mean by positive selection? It's also not clear from your reference.
Fig.2 I hope that the original picture is of high quallity. An explanation is needed on how the tree is built on Fig. 2e;
271 What do you mean by "The unique clusters in the various sequence"?
302- 316 and 322-339 (see above). These data would look more advantageous when compared with at least one other genome. However, I do not insist on this;
379-380 The genome.... is able.....
417-422 Probably this part is from Materials and Methods
420 Plant growth effect can not be detected by photo without sufficient replications and relevant measurements;
Fig.4 This figure combines too heterogeneous material. Sequence alignment probably not needed;
437 phylogenetic relationship ... was developed (not sure, but sounds strange);
445 What is OsNIP2;1?
Fig.5 The effect is not very pronounced, so you need to soften the text a little;
479 I think it's better to remove the numbers from the conclusion.
Round 2
Reviewer 2 Report (Previous Reviewer 2)
Nice work
Author Response
Comments and Suggestions for Authors
Comment: Nice work
Response: Thank you for the appreciation and valuable suggestion during the first round of revision.
Reviewer 3 Report (New Reviewer)
Almost everything is ok. However,
412-416 - at least minimal statistics needed: deviations (+/-), statistical significance of difference (P) etc...
286 - and other - the term "cluster" is not good for description of shared and unique gene groups in Venn diagram
Author Response
Comments and Suggestions for Authors
Comment: Almost everything is ok.
Response: Thank you for the encouraging comment. We have addressed all the comments from both reviewers and revised the MS carefully to rectify possible errors.
Comment: 412-416 - at least minimal statistics needed: deviations (+/-), statistical significance of difference (P) etc...
Response: As per the suggestion we have added standard error in the text and also provided the graph of the Comparison of root, shoot length and fresh weight and dry weight of control vs silicate solubilizing bacteria-treated plants in Figure 4e-h.
Comment: 286 - and other - the term "cluster" is not good for description of shared and unique gene groups in Venn diagram
Response: Agree, with the comment. As per the user manual of the tool “the overlapping cluster means the cluster contains proteins from different species”. So we have now changed the “unique and shared cluster” to “unique and shared proteins”.
This manuscript is a resubmission of an earlier submission. The following is a list of the peer review reports and author responses from that submission.
Round 1
Reviewer 1 Report
This manuscript provides genomic sequences for an isolate of Enterobacter sp. LR6, which appears to be part of the E. cloacae complex based on 16s rRNA phylogenetic analysis. The authors provide a supplemental list of 732 transporter genes, with Table 1 representing an array of stress-related enzymes and transporters found in this species, as well as analysis of transport capabilities of a GlpF aquaglyceroporin. With a large focus on the GlpF pore, the authors did not provide adequate detail on how this species has the potential as a Plant Growth Promoting Bacteria (PGPB), and whether or not the genomic signatures for silicase activity were present.
PGPB is a popular topic in the current literature, looking at the presence or absence of certain proteins and metabolites that aid plants in nutrition or defense responses to abiotic and biotic stress. Two very important genes are those that produce either indole-3-acetic acid (IAA) or 1-aminocyclopropane-1-carboxylic acid deaminase (ACCd). Neither of these were discussed, nor was any strong argument on the involvement of genes for a symbiotic relationship with the plant. Enterobacter spp have been shown to be an endophyte in a number plant species. An example of such analysis can be found in Ramesh et al. (2014).
Ramesh et al. 2014. Plant growth promoting traits in Enterobacter cloacae subsp. dissolvens MDSR9 isolated from soybean rhizosphere and its impact on growth and nutrition of soybean and wheat upon inoculation. Agricultural Research3: 53-66.
The most important oversite is whether or not this species contains any putative silicase genes. Similar to the silicatein from sponges, bacterial species contain different classes of carbonic anhydrase enzymes that not only have the ability to break the C-O bond (as is the case for carbonic acid), but some also have the ability to break the Si-O bond releasing silicic acid from silicate materials. There is literature that describes these responses in bacteria, including a recent publication by Kauret et al. (2022).
Kauret et al. 2022. A novel, simple, and quick plate assay to screen silicolytic bacteria and silicase production using different substrates. Bioresource Technical Reports 17: 1000971.
Also, with this bacterium able to survive in high heat and salinity, the authors did not mention or address any other virulence factors that could make this a potential pathogen to humans and livestock when present in a high concentrations, as would be needed if used as a soil or seed inoculum. E. cloacae complex is a known nosicomal or opportunistic pathogen that has shown infection in hospitalized patients as well as livestock. The concern is the presence of multidrug resistant and other virulence genes that are both intrinsic and present on plasmids (Annavajhala, Gomez-Simmonds & Uhlemann, 2019). In my opinion a risk analysis should be performed when a novel species is identified. Based on the experimental design, the selection pressure was high for bacteria that can live in extreme environments (i.e. the rhizosphere material was dried prior to plating) and this could select for a unique isolate that poses a greater risk as an emerging pathogen compared to the Enterobacter spp community in the rhizosphere as a whole.
Annavajala, Gomez-Simmonds, & Uhlemann. 2019. Multidrug-resistant Enterobacter cloacae complex emerging as a global, diversifying threat. Frontiers in Microbiology 10:44.
Furthermore, with the experimental design, the selection process used is more routinely employed for the detection of unique or rare antimicrobial or beneficial compounds that can be synthesized for use as prophylactic or therapeutic treatments. As stated in the manuscript, the authors suggest that this isolate has the capability of being a PGPB, however, there was no supporting data that show this specific isolate has any effect on plant growth. Genomic data alone cannot define the type of relationship between the host and microorganism (i.e. beneficial, commensal, or pathogen). The density of this particular isolate compared to the other bacteria present within the rhizosphere of the rice was not evaluated and the authors over interpreted the potential importance that this species plays in the growth and defense of plants. Again, genomic data alone does not accurately predict the influence this isolate may or may not have on its environment.
A final general comment refers back to virulence and other genes present on plasmids versus intrinsically present within the chromosomal DNA. The authors did not mention the potential that this bacterium could be harboring plasmids with additional genetic potential, including virulence and silicase elements. The larger than expected genomic size compared to reference strains suggests this is the case. And if plasmid DNA is included, this would significantly impact the whole genome analyses performed and would explain why the 16s rRNA phylogenetic relationship differs from the other, more inclusive analyses.
For this to be a suitable paper, based on the title and authors view of the capabilities of this Enterobacter sp, I propose the following information be added prior to acceptance for publication:
1) The presence or absence of AACd, IAA and other important PGPB genes that would support the interpretation that this bacterium could positively influence plant growth.
2) The presence or absence of carbonic anhydrase genes with discussion on which class they belong to as this will give insight into the silicate weathering potential of this particular isolate.
3) Whether this bacterium can grow at lower pH ranges (i.e. pH >4). This is important to understand its potential to withstand gastric pH in humans and livestock and be useful to understand its potential as an opportunistic pathogen.
4) If the authors have the ability to, the presence or absence of plasmids within this isolate should be stated. It would be important to understand if carbonic anhydrase genes (if present) are intrinsic or plasmid localized. A discussion on the importance of plasmids for Enterobacter spp would also be important as this is what harbors many of the virulence factors mentioned by the authors.
The following are additional, more specific comments.
Page 2
Line 39: Provide reference for the inability of plants to absorb silicates from the soil. As a note, plants do have the capacity to absorb larger polymers and have been shown to absorb silica nanoparticles, indicating there are endocytic mechanisms that may allow them to absorb various forms of silicon, there is just not yet data to support or refute this.
Line 40: Provide reference that silicates are mostly converted to silicic acid by biological activities.
Line 43: What is meant by “combine with the organic compounds”, consider revision of this sentence for clarity.
Line 44: “inhibit the penetration of fungal hyphae” This is an out-dated theory on how silicon was reducing fungal symptoms. There is now data that show silicon treatment does not reduce fungal penetration (or insect probing) and the response is a reduction in branching and growth within the deeper leaf tissue due to changes in the cellular environment. Consider omitting or rephrasing this portion of the sentence.
Line 45: “…is facilitated only with the help of AQPs.” Omit ‘only’. The AQP component of silicon transport in plants is not the only pathway this nutrient may move. NIP2s are only one pore that has been studied, and in addition to the arsB pore complex, there are likely several other mechanisms that have yet to be discovered. By using these definitive descriptive terms, the information is falsely given that these pores have more significance over other cellular mechanisms.
Line 53-54: omit ‘proven’ unless there are multiple references with strong compelling evidence that this is true. Additionally, this is not the most economical approach at this time as our understanding of the rhizosphere dynamics and impact of microbial communities and densities on nutrient mobilization is still in its infancy. Consider omission of this portion of the sentence. Also, include references.
Line 56-57: Is there a previous paper that describes this Enterobacter sp LR6, or is this in reference to the current data presented here? And if the latter, there is no discussion on phosphatase or silicase enzymes identified by this species.
Line 63-64: Provide references for these two sentences.
Line 65-66: This study does not indicate or test that the species have more or less SSB in their rhizosphere compared to other plant species. The study also had inappropriate controls, as sterile media was used there should have been a treatment with A2-1 in order to determine if the enhanced growth was due to the specific isolate or due to having colonization of the roots. Consider using a more relevant study from rice, grain species, or angiosperm on SSB or PGPB.
Line 66-68: This is a continual perpetuation of misguided information on silicon accumulation in plants. The data presented in this manuscript on the % Si in these lower plant divisions is biased and again, outdated. This was from a paper by Takahashi and Miyake (1976) that analyzed silicon content in various plant species to try and understand how ‘silicon accumulators’ evolved. In the 1976 study, there were a total of 175 species tested, 147 angiosperm, 12 gymnosperm, 14 Pteridophyta, and only 2 Bryophyta. Specifically, with Sphagnum cymbifolium warnst containing 1.37% and Marchantia polymorpha L. containing 5.55%. If the angiosperm were limited to testing rice and wheat, its % Si value would be drastically higher than the Bryophyta. This misinterpretation of an incomplete dataset was recently addressed by Thummel (2019) that analyzed phytolith production in Bryophyta and found that only a few select species had the capability to bioaccumulate Si and that this is not a common trait of lower plant species.
Thummel, Brightly & Stromberg. 2019. Evolution of phytolith deposition in modern bryophytes and implications for the fossil record and influence on silica cycle in early land plant evolution. New Phytologist221: 2273-2285.
Takahashi & Miyake.1976. can be found within: Ma & Takashi. 2002. Soil, Fertilizer, and Plant silicon research in Japan. Elsevier, Amsterdam, Netherlands. pp. 64-69; 205-215.
Page 3
Line 78: Please provide information on where this field was located, what soil type was present and any other relevant environmental or physical properties that may influence microbial diversity and density.
Line 80: Include the temperature and drying time used, as this could selectively exclude the viability of other bacterial species from the rhizosphere.
Line 80: How were these isolates maintained prior to testing? Were they subcultured? How old was the culture prior to testing for silicate and phosphate solubilization?
Line 80: How were isolates determined? Was each colony viewed as an independent isolate? And if so, were there differences in colony or cellular morphology among the isolated bacteria?
Page 4
Line 99: Was there an outgroup used for the phylogenetic analysis?
Line 103: ‘The genomic DNA’…. Does this kit have the ability to exclude plasmid DNA, and if so, please indicate the steps that were taken and methods used to ensure only gDNA was isolated. If plasmid DNA may also be present, consider revision of ‘genomic DNA’ to simply ‘DNA’ or indicate that plasmid DNA may also be present.
Line 104: Indicate which Promega kit was used for DNA purification (i.e. Wizard SV).
Line 119: ‘other closely related type strains’. Which strains were used, how many and where is this database located?
Line 120: Provide reference of dDDH and ANI analyses.
Page 5
Line 143: ‘closely related type strain were used’ Indicate strain and any other ID names or markers that would make it easier to find.
Page 6
Line 171: How many times was this assay performed? And how many transformants for each treatment were used (i.e. n=?).
Line 172: Remove ‘silicic acid’. While I agree that GeO2 is a useful proxy for silicic acid transport, the authors are not testing for silicic acid transport, they are inferring based on the GeO2 results.
Page 7
Line 181: 63 isolates seem like a low count for rhizosphere bacteria, is there a reason for this, or is this typical? Were these only the isolates that had SSB activity or the total number from the initial inoculum?
Line 181: Where there any isolates that only had SSB or PSB activity? Possibly include this information if available.
Line 184: Is there a cut-off or interpretation for various zones of inhibition? How does 2 mm compare to other SSB? Would this isolate be considered as having low-, moderate-, or high-solubilizing activity?
Page 8
Figure 1: Why aren’t the Enterobacter spp grouping into similar clades based on the 16s rRNA? Are these complete sequences used for the phylogenetic analysis or trimmed? Please indicate in the materials and methods the OTU used for the analysis.
Page 9
Line 215: … LR6 was found to be closer to E. hormaechei subsp. Xiangfangensis… The relatedness between Enterobacter spp based on genetic analysis is unreliable due to their ability to accumulate and exchange plasmids and other genetic elements (Paauw 2008 & 2009). The larger genomic size suggests that this particular collection of genomic data is from a species harboring plasmid and chromogenic DNA. If the data sets used for ANI and dDDH analysis lacks plasmid elements (i.e. the strains did not contain them or they were removed from the dataset), this would significantly impact the data analysis and make interpretation less reliable.
Paauw et al. 2009. Identification of resistance and virulence factors in an epidemic Enterobacter homaechei outbreak strain. Microbiology 155. 1478-1488.
Paauw et al. 2008. Genomic diversity within Enterobacter cloacae complex. PLOSone 3(8): e3018.
Page 11
Many of these observations from comparative genome analysis indicate the likelihood of plasmid DNA contributing to the unique gene sequences identified. More specifically, the larger genomic size and higher number of unique clusters for LR6. Discussion on plasmid contribution to Enterobacter sp diversity would be helpful.
Page 13
Line 275-278: As the title and abstract of this paper describes the identification of genes involved in plant growth promotion and stress mitigation, the identification of various bacterial transporters is irrelevant. More useful tables would include genes that support endophytic or symbiotic relationships with plants, solubilize various nutrients (especially N,P, and K), those that have roles in antimicrobial or antifungal activities, etc.
Table 1: Is the genes/operon/protein column listing identified genomic elements from the LR6 isolate? And if so, could you provide a supplemental table that has the sequence or location within the deposited genome of these genes?
Tables: Should include additional Tables that show the presence of any putative PGPB or silicase/phosphatase genes.
Page 15
Line 289: Siderophore production is not a strict characteristic of beneficial microbes. This protein chelates Fe and limits its presence in the rhizosphere, thus reducing the nutrient availability to other microbes. It does not supplement plants with Fe. Many soil microbes produce these molecules and its presence in the LR6 isolate is not compelling evidence of its presence as a PGPB or beneficial microbe.
Page 16
Line 302: Omit the last portion of this sentence from … as it encodes for the various proteins to the end. There are three processes that are discussed further and only stress mitigation is mentioned here.
Additionally in this section, the authors are using genomic data to infer environmental survival and function. There is no indication on how common this isolate is in the natural environment. Based on the experimental design, it is more likely that this is a rare isolate, which makes it more difficult to accurately predict its involvement without performing additional studies. Genomic data alone cannot support functionality. These are common genes among Enterobacteriaceae members and not unique to this isolate, more notably, being present in the true pathogen Salmonella enteritica subsp. typhi further exemplifies that its presence does not indicate plant growth promotion or beneficial contributions of this isolate to the rhizosphere.
It might be more accurate to describe this section as ‘Genetic Potential of Enterobacter sp LR6 for Survival in plant rhizosphere’ as there are no genes discussed that would indicate growth promotion.
This section contains a long list of genes identified in this species. This information with genomic location or sequences in a Table would be useful.
Page 17
Section 3.8 There has been no data to support that this isolate has any growth-promoting activity. It is a great overreach to state that this is tied to the presence of phosphate solubilizing genes. Consider rephrasing this section to indicate the potential for growth-promoting activity.
Additionally, the sequestration of Fe does not promote growth, but gives this isolate the potential edge to live in a nutrient-limiting environment.
Components that support plant growth (i.e. hormones, metabolites, or other nutrient solubilizing components) would be more appropriate to add to this section to support the genetic potential present within this isolate for plant growth promotion.
Page 18
Line 377: …. empty vector grew well… Replace ‘well’ with a better descriptive term (i.. grew at a higher density).
The annotated features for the GlpF are described here, but it may be useful to share the predicted protein sequence as well (or provide this in a supplemental file).
Figure 4c: Indicate the serial dilution of the colonies below the images. Also, for GeO2, the serial dilutions look close to one-another. How many biological replicates were performed with this assay and was it done more than a single time? Please indicate this is the figure legend. A quantitative analysis of the colony density would be helpful to determine the sensitivity of the transformed yeast to these toxic compounds.
Page 20
Line 405-406: “This study emphasized mainly how this silicate solubilizing bacteria act as plant growth-promoting bacteria, their movement towards the rhizosphere, its colonization, plant growth promotion, and Si uptake.”
This manuscript demonstrated none of these things. It did not even provide evidence that this isolate was beneficial and could be argued to be pathogenic with the genetic factors listed, as well as its ability to survive at elevated temperatures and lower pH.
Line 407: “It is also responsible for the transportation of Si, which has been confirmed by the yeast assay.” This is not true. It is inferred to transport Si based on the ability to transport Ge. Again, I do agree that Ge can indicate Si transport, but it does not confirm this capability, it only supports the probability that it can. Consider revision of this sentence to reflect this.
Line 409: “Therefore, LR6 can provide a sustainable solution to improve crop yield.” Again, there is no evidence that this isolate has any impact on yield or that it has the ability to colonize the rice roots to have a significant impact on the overall health and productivity of the plant.
Line 411:…. Highlighted genes involved in silicate solubilization…. No genes on this were identified, only a bacterial transporter that would reduce silicic acid in the environment as the silicon would move into the bacterial cell.
Line 412-413: I disagree with this last statement, as the genomic information is useful for data mining, it does not provide any insight into molecular mechanisms. The phenotypic traits and environmental role of this isolate were not presented, and thus no conclusions can be drawn from genomic data other than the presence or absence of genes.
Reviewer 2 Report
In this study, authors isolated strains from rice root and found the strain LR6 can solubilize silica in culturing medium. Then they completed the genome sequencing and comparative genomic analysis. The methods are acceptable. Whereas, there are tow major concerns below. The writing needs largely editing. Several examples are listed below.
Major point:
1. The LR6 was found to solubilize silica optimally in Bunt and Rovira media. Whereas, authors also should prove the strain can solubilize silica in rice root and can largely amount habit in the root area. Otherwise, the utilization potentiality will low and the significant will less.
2. AQPZ and one GlpF were detected in the genome. The value point is to explore the evolution relationship of these two genes between LR6 and others, which will decipher the origin of these two gene, and reveal that why some species containing them and some not, whether the solubilize silica phenotype of bacterial species associating with these two genes?
Minor points
L18, L26: With or without blank before %? Thoroughly check the manuscript.
L18: It is better adding the Thousands separator.
L65: First appearance of SSB. Full name should be given.
L179: Italic Enterobacter. Thoroughly check the manuscript.
All Latin names in Figures should be italic.
Figure 3. The (a) actually contains three panels and also the (b), which should be separately marked and added legends.
Round 2
Reviewer 2 Report
Major points were not directly addressed. Minor points were unsolved.